# TRIPLE-SEARCH: DIFFERENTIABLE JOINT-SEARCH OF NETWORKS, PRECISION, AND ACCELERATORS

## ABSTRACT

The record-breaking performance and prohibitive complexity of deep neural networks (DNNs) have ignited a substantial need for customized DNN accelerators which have the potential to boost DNN acceleration efficiency by orders-of-magnitude. While it has been recognized that maximizing DNNs' acceleration efficiency requires a joint design/search for three different yet highly coupled aspects, including the networks, adopted precision, and their accelerators, the challenges associated with such a joint search have not yet been fully discussed and addressed. First, to jointly search for a network and its precision via differentiable search, there exists a dilemma of whether to explode the memory consumption or achieve sub-optimal designs. Second, a generic and differentiable joint search of the networks and their accelerators is non-trivial due to (1) the discrete nature of the accelerator space and (2) the difficulty of obtaining operation-wise hardware cost penalties because some accelerator parameters are determined by the whole network. To this end, we propose a Triple-Search (TRIPS) framework to address the aforementioned challenges towards jointly searching for the network structure, precision, and accelerator in a differentiable manner, to efficiently and effectively explore the huge joint search space. Our TRIPS addresses the first challenge above via a heterogeneous sampling strategy to achieve unbiased search with constant memory consumption, and tackles the latter one using a novel co-search pipeline that integrates a generic differentiable accelerator search engine. Extensive experiments and ablation studies validate that both TRIPS generated networks and accelerators consistently outperform state-of-the-art (SOTA) designs (including co-search/exploration techniques, hardware-aware NAS methods, and DNN accelerators), in terms of search time, task accuracy, and accelerator efficiency. All codes will be released upon acceptance.

## 1 INTRODUCTION

The powerful performance and prohibitive complexity of deep neural networks (DNNs) have fueled a tremendous demand for efficient DNN accelerators which could boost DNN acceleration efficiency by orders-of-magnitude (Chen et al., 2016). In response, extensive research efforts have been devoted to developing DNN accelerators. Early works decouple the design of efficient DNN algorithms and their accelerators. On the algorithms level, pruning, quantization, or neural architecture search (NAS) are adopted to trim down the model complexity; On the hardware level, various FPGA-/ASIC-based accelerators have been developed to customize the *micro-architectures* (e.g., processing elements dimension, memory sizes, and network-on-chip design) and algorithm-to-hardware *mapping methods* (e.g., loop tiling strategies and loop orders) in order to optimize the acceleration efficiency for a given DNN. Later, hardware-aware NAS (HA-NAS) has been developed to further improve DNNs' acceleration efficiency for different applications (Tan et al., 2019).

More recently, it has been recognized that (1) optimal DNN accelerators require a joint consideration/search for all the following different yet coupled aspects, including DNNs' network structure, the adopted precision, and their accelerators' micro-architecture and mapping methods, and (2) merely exploring a subset of these aspects will lead to sub-optimal designs in terms of hardware efficiency or task accuracy. For example, the optimal accelerators for networks with different structures (e.g., width, depth, and kernel size) can be very different; while the optimal networks and their bitwidths

for different accelerators can differ a lot (Wu et al., 2019). However, the direction of jointly designing or searching for all the three aspects has only been slightly touched on previously. For example, (Chen et al., 2018; Gong et al., 2019; Wang et al., 2020) proposed to jointly search for the structure and precision of DNNs for a fixed target hardware; (Abdelfattah et al., 2020; Yang et al., 2020; Jiang et al., 2020a;b) made the first attempt to jointly search for the networks and their accelerators, yet either their network or accelerator choices are limited due to the prohibitive time cost required by their adopted reinforcement learning (RL) based methods; and EDD (Li et al., 2020) contributed a pioneering effort towards this direction by formulating a differentiable joint search framework, which however only consider one single accelerator parameter (i.e., parallel factor) and more importantly, has not yet fully solved the challenges of such joint search.

Although differentiable search is one of the most promising ways in terms of search efficiency to explore the huge joint search space as discussed in Sec. 4.2, plethora of challenges exist to achieve an effective generic joint search for the aforementioned three aspects. First, Challenge 1: to jointly search for a network and its precision via differentiable search, there exists a dilemma whether to activate all the paths during search. On one hand, the required memory consumption can easily explode and thus constrain the search's scalability to more complex tasks if all paths are activated; on the other hand, partially activating a subset of the paths can lead to a sequential training of different precision on the same weights, which might result in inaccurate accuracy ranking among different precision as discussed in (Jin et al., 2020). Second, Challenge 2: the accelerators' parameters are not differentiable, and it is non-trivial to derive the operation-wise hardware-cost penalty in order to perform differentiable search (in considering search efficiency). This is because the optimal accelerator is often determined by the whole network instead of one specific operation/layer due to the fact that some accelerator parameters (e.g., the loop order) need to be optimized for the whole network.

In this paper, we aim to address the aforementioned challenges towards scalable generic joint search for the network, precision, and accelerator. Specifically, we make the following contributions:

- We propose a Triple-Search (TRIPS) framework to jointly search for the network, precision, and accelerator in a differentiable manner to efficiently explore the huge joint search space which cannot be afforded by previous RL-based methods due to their required prohibitive search cost. TRIPS identifies and tackles the aforementioned challenges towards scalable generic joint search of the three for maximizing both the accuracy and acceleration efficiency.

- We develop a *heterogeneous* sampling strategy for simultaneous updating the weights and network structures to (1) avoid the need to sequentially train different precision and (2) achieve unbiased search with constant memory consumption, i.e., solve the above Challenge 1. In addition, we develop a novel co-search pipeline that integrates a differentiable hardware search engine to address the above Challenge 2.

- Extensive experiments and ablation studies validate the effectiveness of our proposed TRIPS framework in terms of the resulting search time, task accuracy, and accelerator efficiency, when benchmarked over state-of-the-art (SOTA) co-search/exploration techniques, HA-NAS methods, and DNN accelerators. Furthermore, we visualize the searched accelerators by TRIPS to provide insights towards efficient DNN accelerator design in Appendix.

## 2 RELATED WORKS

**Hardware-aware NAS.** Hardware-aware NAS has been proposed to automate the design of efficient DNNs. Early works (Tan et al., 2019; Howard et al., 2019; Tan & Le, 2019) utilize RL-based NAS that requires a massive search time/cost, while recent works (Wu et al., 2019; Wan et al., 2020; Cai et al., 2018; Stamoulis et al., 2019) explore the design space in a differentiable way (Liu et al., 2018) with much improved searching efficiency. Along another direction, one-shot NAS methods (Cai et al., 2019; Guo et al., 2020; Yu et al., 2020) pretrain the supernet and directly evaluate the performances of the sub-networks in a weight sharing manner as a proxy of their independently trained performances at the cost of a longer pretrain time. In addition, NAS has been adopted to search for quantization strategies (Wang et al., 2019; Wu et al., 2018; Cai & Vasconcelos, 2020; Elthakeb et al., 2020) for trimming down the complexity of a given DNN. However, these works leave unexplored the

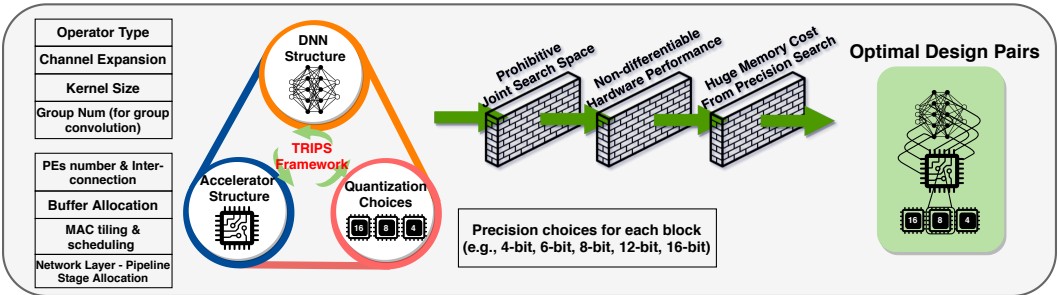

Figure 1: Illustrating our TRIPS framework: the large joint space and tackled challenges.

hardware design space, which is a crucial enabler for DNN's acceleration efficiency, thus can lead to sub-optimal solutions.

**DNN accelerators.** Motivated by customized accelerators' large potential gains, SOTA accelerators (Du et al., 2015; Chen et al., 2017) innovate micro-architectures and algorithm-to-hardware mapping methods to optimize the acceleration efficiency, given a DNN and the hardware specifications. However, it is non-trivial to design an optimal accelerator as it requires cross-disciplinary knowledge in algorithm, micro-architecture, and circuit design. SOTA accelerator design relies on either experts' manual design, which is very time consuming or design flow (Chen et al., 2005; 2009; Rupnow et al., 2011) and DNN accelerator design automation (Wang et al., 2016; Zhang et al., 2018a; Guan et al., 2017; Venkatesan et al., 2019; Wang et al., 2018a; Gao et al., 2017). As they merely explore the accelerator design space, they can result in sub-optimal solutions as compared to SOTA co-search/exploration methods and our TRIPS framework.

**Co-exploration/search techniques.** Pioneering efforts have been made towards jointly searching of DNNs and their accelerators to some extent. For joint searching of DNNs and their precision, (Chen et al., 2018; Gong et al., 2019; Wang et al., 2020) adopt either differentiable or evolutionary algorithms yet without exploring their hardware accelerators. For joint searching of DNNs and their accelerators, (Abdelfattah et al., 2020; Yang et al., 2020; Jiang et al., 2020a;b) conduct RL-based search for the networks and some accelerator parameters/templates, where they strictly constrain the search space of the network or accelerator to achieve a practical RL search time, limiting their scalability and achievable efficiency. (Lin et al.) is another pioneering work which co-designs the newtork and accelerator in a sequential manner based on the fact that the accelerator's design cycle is longer than the networks. EDD (Li et al., 2020) extends differentiable NAS to search for layer-wise precision and the accelerators' parallel factor, which is most relevant to our TRIPS. EDD has not yet fully solved the joint search challenges. First, it does not discuss or address the potentially explosive memory consumption issue of such joint search; second, EDD's accelerator search space only includes the parallel factor, which can be strictly limited to their accelerator template and cannot generalize to include common accelerator parameters such as the tiling strategies.

Built upon prior art, our TRIPS targets a scalable generic joint search framework to optimally search for the network, its precision, and adopted accelerator in a differentiable manner for improving efficiency.

# 3 THE PROPOSED TECHNIQUES

In this section, we describe our proposed techniques for enabling TRIPS, where Sec. 3.1 provides TRIPS's formulation, Sec. 3.2 and Sec. 3.3 introduce TRIPS's enablers that address the key challenges of scalable generic joint search for networks, precision, and accelerators, and Sec. 3.4 unifies the enablers to build a comprehensive co-search framework.

## 3.1 TRIPS: FORMULATION

Fig. 1 shows an overview of TRIPS, which jointly searches for the networks (e.g., kernel size, channel expansion, and group number), precision (e.g., 4-/6-/8-/12-/16-bit), and the accelerators (e.g., PE array type, buffer size, and tiling strategies of each memory hierarchy) in a differentiable manner.

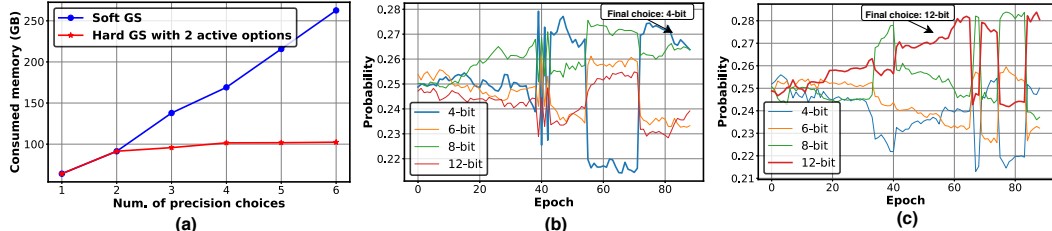

Figure 2: (a) GPU memory consumption comparison between soft Gumbel Softmax (GS) and hard GS sampling (two active choices) for precision search. Probability of each precision choice during the search process in the 4-th block when searching with: (b) hard GS sampling for updating both weights $\omega$ and precision choices $\beta$ (result in the lowest 4-bit), and (c) the proposed heterogeneous sampling for updating $\omega$ and $\beta$ (result in the highest 12-bit).

TRIPS targets a *scalable* yet *generic* joint search framework, which we formulate as a bi-level optimization problem:

$$\min_{\alpha, \beta} \ L_{val}(\omega^*, net(\alpha), prec(\beta)) + \lambda L_{cost}(hw(\gamma^*), net(\alpha), prec(\beta)) \quad (1)$$

$$s.t. \quad \omega^* = \arg\min_{\omega} L_{train}(\omega, net(\alpha), prec(\beta)), \quad (2)$$

$$s.t. \quad \gamma^* = \arg\min_{\gamma} L_{cost}(hw(\gamma), net(\alpha), prec(\beta)) \quad (3)$$

Where $\alpha$, $\beta$, and $\gamma$ are the continuous variables parameterizing the probability of different choices for the network operators, precision bitwidths, and accelerator parameters as in (Liu et al., 2018), $\omega$ is the supernet weights, $L_{train}$, $L_{val}$, and $L_{cost}$ are the loss during training and validation and the hardware-cost loss, and $net(\alpha)$, $prec(\beta)$, and $hw(\gamma)$ denote the network, precision, and accelerator characterized by $\alpha$, $\beta$, $\gamma$, respectively.

## 3.2 TRIPS Enabler 1: Heterogeneous sampling for precision search

As discussed in Sec.1, there exists a dilemma (i.e., memory consumption explosion or biased search) whether to activate all the paths during precision search, for addressing which we have developed a heterogeneous sampling. Next, we first use real experiments to illustrate the joint search dilemma, and then introduce our heterogeneous sampling which effectively address those challenges/issues.

**Activating all choices - memory explosion and entangled correlation among choices.** During precision search, activating all the precision choices as (Wu et al., 2018; Gong et al., 2019) can easily explode the memory consumption especially when the precision is co-searched with the network structures. While composite convolutions (Cai & Vasconcelos, 2020) for mixed precision search can mitigate this memory explosion issue during search by shrinking the required computation, yet this large memory consumption issue would still exist during training when updating the precision parameters, i.e., $\beta$ in Eq. (1). For example, as shown in Fig. 2 (a), the measured GPU memory consumption of co-searching the network and precision on ImageNet grows linearly with the number of precision choices if activating all precision choices during search. In addition, the entangled correlation (e.g., co-adaptation (Hong et al., 2020), correlation (Li et al., 2019), and cooperation (Tian et al., 2020)) among different precision choices leads to a large gap between the supernet during search and the final derived network, thus failing the joint search.

**Activating only a subset of choices - Biased search.** For addressing the above issues of memory explosion and correlation among choices, one natural choice is to adopt hard Gumbel Softmax by reducing the memory consumption, which however can lead to a biased search and thus poor performance. Specifically, activating only a subset of the precision choices implies a sequential training of different precisions that can lead to inaccurate performance

Table 1: Comparing the resulting accuracy when training a fixed network using different precision schedules, where high2low and low2high denote sequential training from high precision to low precision and the inverse case, respectively.

| Strategy | Acc | | | | |
|---|---|---|---|---|---|
| | 4-bit (%) | 8-bit (%) | 12-bit (%) | 16-bit (%) | 32-bit (%) |
| Independent | 63.52 | 67.44 | 67.56 | 67.65 | 68.21 |
| high2low | 59.29 | 45.09 | 45.45 | 45.15 | 65 |
| low2high | 4.36 | 26.55 | 43.58 | 63.3 | 63.5 |
| joint | 63.28 | 66.98 | 67.21 | 67.23 | 67.36 |

ranking. This is because a sequential training means different precision choices are applied on top of the same weights and activations. As a result, different precision choices can interfere with each other and different training order would lead to a different result. For better understanding, we next show two concrete experiments.

Co-search network and precision using hard Gumbel Softmax: Fig. 2 (b) shows the resulting precision probability evolution when co-searching the network and precision on CIFAR-100 using hard Gumbel Softmax (activating two precision choices) without imposing any hardware-cost constraints, indicating the desired precision choice would be the highest precision. However, as shown in Fig. 2 (b), the block co-searched using hard Gumbel Softmax collapse to the lowest precision (i.e., the highest probability towards the end of the search is the lowest precision choice 4-bit), indicating an ineffective search direction. Note that the fluctuation in the probability of different precision choices is caused by the intermittent activation of the block due to the hard Gumbel Softmax sampling.

Sequential training of a fixed network with multiple precision choices: As observed in (Jin et al., 2020), when training a fixed network with multiple precision choices, either ascending or descending the precision will incur an inferior convergence and thus chaotic accuracy ranking among different precision choices. For example, as shown in Tab. 1, we compare the accuracy of a fixed network (all blocks adopt the k3e1 (kernel size 3 and channel expansion 1) structure in (Wu et al., 2019)) under different precision choices, when being trained with different precision schedules, and find that only jointly training all the precision choices can maintain the ranking consistent with that of independently trained ones, while sequential training leads to both inferior accuracy and ranking.

**Proposed solution - Heterogeneous sampling.** To tackle both aspects of the aforementioned dilemma, we propose a heterogeneous sampling strategy as formulated in Eq. (4) where $\bar{W}^l$ / $\bar{A}^l$ are the composite weights / activations of the $l$-th layer as in (Cai & Vasconcelos, 2020) which are the weighted sum of weights / activations under different precision choices, e.g., $W_j^l$ is the weights quantized to the $j$-th precision among the total $J$ options of the $l$-th layer. In our heterogeneous sampling, for updating the weights in Eq. (2), we jointly update the weights under all the precision choices weighted by their corresponding soft Gumbel Softmax $GS(\beta_j^l)$, where $\beta_j^l$ parameterizes the probability of the $j$-th precision in the $l$-th layer, and the gradients can be estimated by STE (Zhou et al., 2016) as $\partial L_{train}/\partial A^l \approx \partial L_{train}/\partial \bar{A}^l$ so that no extra intermediate feature maps are needed to be stored into the memory during backward. For updating $\beta$, we adopt hard Gumbel Softmax (Jang et al., 2016) with one-hot outputs $GS_{hard}(\beta_j^l)$ to save memory and computation while reducing the correlation among precision choices. In the same co-search setting as Fig. 2 (b), all the blocks searched using our proposed heterogeneous sampling converge to the highest precision choice towards the end of the search as shown in Fig. 2 (c).

$$A^{l+1} = \bar{W}^l * \bar{A}^l = \sum_{j=1}^{J} \bar{\beta}_j^l W_j^l * \sum_{j=1}^{J} \bar{\beta}_j^l A_j^l \;\; where \; \bar{\beta}_j^l = \begin{cases} GS(\beta_j^l) & \text{if updating weight} \\ GS_{hard}(\beta_j^l) & \text{if updating } \beta \end{cases} \quad (4)$$

### 3.3 TRIPS ENABLER 2: DIFFERENTIABLE ACCELERATOR SEARCH ENGINE

**Motivation.** Although EDD (Li et al., 2020) also co-searches the accelerator with the network, their search space is limited to include only the parallel factor within their template which can be analytically fused into their theoretical computational cost, whereas this is not always applicable to other naturally non-differentiable accelerator design parameters such as tiling strategies. A more general and efficient search engine is needed towards generic differentiable accelerator search.

**Search algorithm.** We propose a differentiable search engine to efficiently search for the optimal accelerator (including the micro-architectures and mapping methods) given a DNN model and its precision based on single-path sampling as discussed in Sec. 3.1. We solve Eq. (3) in a differentiable way:

$$\gamma^* = \arg\min_{\gamma} \sum_{m=1}^{M} GS_{hard}(\gamma^m) L_{cost}(hw(\{GS_{hard}(\gamma^m)\}), net(\{O_{fw}^l\}), prec(\{B_{fw}^l\})) \quad (5)$$

where $M$ is the number of accelerator design parameters. Given the network $net(\{O_{fw}^l\})$ and precision $prec(\{B_{fw}^l\})$, where $O_{fw}^l$ and $B_{fw}^l$ are the only operator and precision activated during forward as discussed in Sec. 3.4, our search engine utilizes hard Gumbel Softmax $GS_{hard}$ sampling

on each design parameter $\gamma^m$ to build an accelerator $hw(\{GS_{hard}(\gamma^m)\})$ and penalize each sampled accelerator parameter with the overall hardware-cost $L_{cost}$ through relaxation in a gradient manner.

**Hardware template.** We adopt a unified template for both the FPGA and ASIC accelerators, which is a parameterized chunk-based pipeline micro-architecture inspired by (Shen et al., 2017). In particular, the hardware/micro-architecture template comprises multiple sub-accelerators (i.e., chunks) and executes DNNs in a pipeline fashion. Each chunk is assigned with multiple but not necessarily consecutive layers which are executed sequentially within the chunk. Similar to Eyeriss, each chunk consists of levels of buffers/memories (e.g., on-chip buffer and local register files) and processing elements (PEs) to facilitate data reuses and parallelism with searchable design knobs such as PE interconnections (i.e., Network-on-chip), allocated buffer sizes, MAC operations' scheduling and tiling (i.e., dataflows), and so on (see more details in Appendix B).

**Discussion about the general applicability.** Our search approach is general and can be applicable to different hardware architectures, since we do not hold any prior assumptions about the adopted hardware architecture. Specifically, for any target hardware architecture, including TPU-like or GEMM or other accelerators, our search approach can be directly applied once given (1) a simulator to estimate the hardware cost, and (2) a set of user-defined searchable design knobs abstracted from the target hardware architecture.

## 3.4 TRIPS: The overall Co-search Framework

**Objective and challenges.** TRIPS's iterative search starts from updating both the supernet weights $\omega$ and accelerator parameters $\gamma$, given the current network $net(\alpha)$ quantized using precision $prec(\beta)$, and then updates $\alpha$ and $\beta$ based on the derived optimal weights $\omega^*$ and accelerator $hw(\gamma^*)$ resulting from the previous step. Updating $\omega^*$ and $\gamma^*$ have been discussed in Sec. 3.2 and Sec. 3.3, respectively. **The key objective** of TRIPS is captured by Eq. (1) which involves all the three major aspects towards efficient DNN accelerators. **The key challenges** in achieving TRIPS include **(1)** the prohibitively large joint search space (e.g., 2.3E+21 in this work) which if not addressed will limit the scalability of TRIPS to practical yet complex tasks; **(2)** the entangled co-adaptation (Hong et al., 2020), correlation (Li et al., 2019), and cooperation (Tian et al., 2020) issues among different network and precision choices can enlarge the gap between the supernet during search and the final derived network, thus failing the joint search; and **(3)** the non-triviality of deriving hardware-cost for the layer/block-wise update during network search, as the hardware-cost is determined by the whole network.

$$Forward: A^{l+1} = \sum_{i=1}^{N} GS_{hard}(\alpha_i^l) O_i(A^l) = O_{fw}^l(A^l) \tag{6}$$

$$Backward: \frac{\partial L_{val}}{\partial \alpha_i^l} = \sum_{k=1}^{K} \frac{\partial L_{val}}{\partial GS(\alpha_k^l)} \frac{\partial GS(\alpha_k^l)}{\partial \alpha_i^l} = \frac{\partial L_{val}}{\partial A^{l+1}} \sum_{k=1}^{K} O_k^l(A^l) \frac{\partial GS(\alpha_k^l)}{\partial \alpha_i^l} \tag{7}$$

$$\frac{\partial L_{cost}}{\partial \alpha_i^l} = \mathbb{1}(GS_{hard}(\alpha_i^l) = 1) L_{cost}^{\alpha_i^l}(hw(\gamma^*), net(\alpha_i^l), prec(\beta)) \tag{8}$$

**TRIPS implementation.** To tackle the three aforementioned challenges, TRIPS integrates a novel co-search pipeline which can be illustrated using the co-search for $\alpha$ as follows and is similarly applicable to co-search for $\beta$ . Note that here we define path to be one of the paralleled candidate operators between the layer input and layer output within one searchable layer, which can be viewed as a coarse-grained (layer-wise) version of the path definition in (Wang et al., 2018b; Qiu et al., 2019).

Single-path forward: For updating both $\alpha$ (see Eq. (6)) and $\beta$ during forward, TRIPS adopts hard Gumbel Softmax sampling (Hu et al., 2020a), i.e., only the choice with the highest probability will be activated to narrow the gap between the search and evaluation thanks to the single-path property of hard Gumbel Softmax sampling. In Eq. (6), $A^l$ and $A^{l+1}$ denote the feature maps of the $l$-th and $(l + 1)$-th layer, respectively, $N$ is the total number of operator choices, $O_i^l$ is the $i$-th operator in the $l$-th layer parameterized by $\alpha_i^l$, and $O_{fw}^l$ is the only operator activated during forward.

Multi-path backward: For updating both $\alpha$ (see Eq. (7)) and $\beta$ during backward, TRIPS activates multiple paths to calculate the gradients of $\alpha$ and $\beta$ through Gumbel Softmax relaxation in order to balance the search efficiency and stability motivated by (Cai et al., 2018; Hu et al., 2020b). For

example, $\alpha_i^l$'s gradients are calculated using Eq. (7), where $K$ is the number of activated choices with the top $K$ Gumbel Softmax probability. Similar to (Cai et al., 2018), $K \in (1, N)$ in TRIPS to control the computational cost.

Hardware-cost penalty: The network search in Eq. (1) is performed in a layer/block-wise manner as in (Liu et al., 2018), thus requiring layer/block-wise hardware-cost penalty which is determined by the layer/block-to-accelerator *mapping method* and the corresponding layer/block execution cost on the optimal accelerator $hw(\gamma^*)$. The optimal mapping method of an accelerator is yet determined by the whole network. To handle this gap, we derive the layer/block-wise hardware-cost assuming that the single-path network derived from the current forward would be the final derived network, as this single-path network has a higher if not the highest probability to be finally derived. In Eq. (8), $\mathbb{1}$ is an indicator denoting whether $\alpha_i^l$ (i.e., the $i$-th operator in the $l$-th layer) is activated during forward.

# 4 EXPERIMENT RESULTS

## 4.1 EXPERIMENT SETUP

**Software settings.** Search space and hyper-params. We adopt the same search space in (Wu et al., 2019) for the ImageNet experiments and disable the first two down sampling operations for the CIFAR-10/100 experiments. We use [4, 6, 8, 12, 16] as candidate precision choices and one block shares the same precision of weights and activations for more hardware friendly implementation. We activate two paths during backward, i.e., $K = 2$ in Eq. (7), for search efficiency. For $L_{cost}$ in Eq. ( 3), we use latency for FPGA as the target metric is Frame-Per-Second (FPS), and Energy-Delay-Product (EDP) for ASIC. **Detailed search and training settings are elaborated in Appendix A.**

Baselines. We mainly benchmark over four kinds of SOTA baselines: (1) the most relevant baseline EDD (Li et al., 2020) which co-searches networks, precision, and accelerators, (2) SOTA methods co-exploring networks and accelerators including HS-Co-Opt (Jiang et al., 2020b), NASAIC (Yang et al., 2020), and BSW (Abdelfattah et al., 2020), (3) SOTA methods co-searching the networks and precision including APQ (Wang et al., 2020) and MP-NAS (Gong et al., 2019), and (4) hardware-aware NAS with uniform precision, including FBNet (Wu et al., 2019), ProxylessNAS (Cai et al., 2018), Single-Path NAS (Stamoulis et al., 2019), and EfficientNet-B0 (Tan & Le, 2019).

**Hardware settings.** To evaluate the generated network and accelerator designs, for FPGA cases, we adopt the standard Vivado HLS (Xilinx Inc., a) design flow, on the target Xilinx ZC706 development board (Xilinx Inc., b), which has a total 900 DSP48s (Digital Signal Processor) and 19.1Mb BRAM (Block RAM). For ASIC implementations, we use the SOTA energy estimation tool Timeloop (Parashar et al., 2019) and Accelergy, (Wu et al., 2019), to validate our generated design's performance, with CACTI7 (Balasubramonian et al., 2017) and Aladdin (Shao et al., 2014) at a 32nm CMOS technology as unit energy and timing cost plugins. **Details about the accelerator search space are discussed in Appendix B.**

## 4.2 BENCHMARK SEARCH EFFICIENCY

To evaluate the superiority of TRIPS in terms of search efficiency, we compare the search space size and search time of TRIPS with both RL-based co-search works and one-shot NAS methods using the reported data from the baselines' original papers as shown in Tab. 2. We can see that TRIPS consistently require notably less search time while handling the largest joint search space on all the considered tasks. In particular, compared with the one-shot NAS methods (Guo et al., 2020; Cai

Table 2: Search efficiency benchmark of TRIPS over co-search works and one-shot NAS methods.

| Method | Dataset | Network Space | Accelerator Space | Precision Space | Joint Space | Search Time (GPU hours) |
|---|---|---|---|---|---|---|
| HS-Co-Opt (Jiang et al., 2020b) | CIFAR-10 | 1.15E+18 | - | - | 1.15E+18 | 103.9 |
| **TRIPS** | CIFAR-10 | 9.85E+20 | 2.24E+27 | 2.40E+15 | **5.30E+63** | **6** |
| BSW (Abdelfattah et al., 2020) | CIFAR-100 | 4.20E+05 | 8.64E+03 | - | 3.63E+09 | 5184 |
| **TRIPS** | CIFAR-100 | 9.85E+20 | 2.24E+27 | 2.40E+15 | **5.30E+63** | **12** |
| HS-Co-Opt (Jiang et al., 2020b) | ImageNet | 2.22E+18 | - | - | 2.22E+18 | 266.8 |
| Once-For-All (Cai et al., 2019) | ImageNet | 2.00E+19 | - | - | 2.00E+19 | 1200 |
| APQ (Wang et al., 2020) | ImageNet | 1.00E+35 | - | 1.00E+10 | 1.00E+45 | 2400 |
| Single One-shot (Guo et al., 2020) | ImageNet | 7.00E+21 | - | - | 7.00E+21 | 288 |
| **TRIPS** | ImageNet | 9.85E+20 | 2.24E+27 | 2.40E+15 | **5.30E+63** | **80** |

et al., 2019) which can be potentially extended to co-search frameworks while suffering from a large pretraining cost, TRIPS achieves 3.6× ~ 30× less search time on ImageNet, while being end-to-end, justifying our choice of differentiable co-search.

## 4.3 BENCHMARK OVER SOTA METHODS

**Co-exploration of networks, precision, and accelerators.** We benchmark our TRIPS framework with SOTA efficient DNN solutions on ImageNet and FPGA-based accelerators under the 512 DSP limits in Fig. 3 following (Abdelfattah et al., 2020). Specifically, we provide four searched results of our TRIPS framework; we use the reported results for EDD, and search for the optimal accelerator in our accelerator space for APQ, MP-NAS, and SOTA hardware-aware NAS methods; for EfficientNet-B0, we apply the SOTA mixed precision strategy searched by (Habi et al., 2020) for a fair comparison and the ProxylessNAS-8bit is reported by APQ (Wang et al., 2020); and the other baselines are all quantized to 8-bit for hardware measurement and the accuracies are from the original papers with-

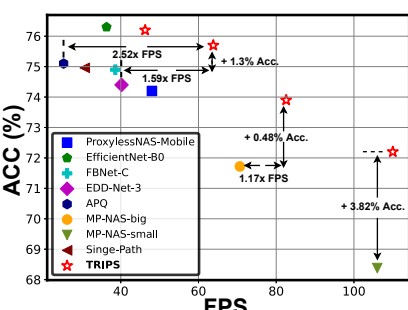

Figure 3: Accuracy vs. FPS trade-off of TRIPS against SOTA efficient DNN solutions on ImageNet.

out considering the quantization effect. We can observe from Fig. 3 that (1) the searched networks by our TRIPS framework consistently push forward the frontier of accuracy-FPS trade-offs, (2) compared with the most relevant baseline EDD, we achieve a +1.3% higher accuracy with a 1.59× FPS. The effectiveness of TRIPS over various SOTA methods that represent most of the existing co-design directions verifies the necessity and effectiveness of co-searching all the three aspects towards efficient DNN accelerators.

**Co-exploration of networks and accelerators.** Software-Hardware co-design is a significant property of our TRIPS framework, so we further benchmark it with both searched precision and fixed-precision over SOTA network/accelerator co-search works for a fair comparison.

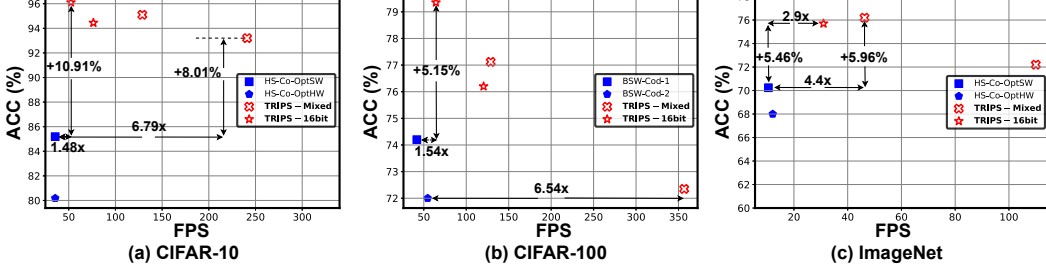

Figure 4: Benchmark TRIPS w/ and w/o precision search (denoted as TRIPS-Mixed and TRIPS-16bit, respectively) with SOTA network/accelerator co-exploration methods (Jiang et al., 2020b; Abdelfattah et al., 2020) on CIFAR-10/100/ImageNet.

Co-search on FPGA. We benchmark with HS-Co-Opt (Jiang et al., 2020b) and BSW (Abdelfattah et al., 2020) on ZC706 under the same DSP limits as the baselines on CIFAR-10/100/ImageNet. Note that all our baselines here adopt a 16-bit fixed-point design, so we provide TRIPS with fixed 16-bit in addition to the one with searched precision for a fair comparison. From Fig. 4, we can see that (1) on both

Table 3: Comparing the accuracy and ASIC efficiency (i.e., EDP and area) of TRIPS and SOTA co-exploration ASIC works (Yang et al., 2020).

| Optimization Methods | Accuracy (%) | EDP (J*clock-cycle) | Area (um²) |
|---|---|---|---|
| NAS → ASIC | 94.17 | 3.30E+06 | 4.83E+09 |
| ASIC → HW-NAS | 92.53 | 2.81E+06 | 3.86E+09 |
| NASAIC | 92.62 | 1.62E+06 | 3.34E+09 |
| **TRIPS** | **94.34** | **4.36E+03** | **5.92E+05** |

CIFAR-10/100 dataset, TRIPS with fixed 16-bit consistently achieves a better accuracy (up to 10.91% and 5.15%, respectively) and higher FPS (up to 2.21× and 2.15×, respectively) under the same DSP constraint, and (2) when co-searching the precision, our method can more aggressively push forward the FPS improvement (up to 6.79× and 6.54×, respectively on CIFAR-10/100), implying the importance of the co-exploration of the precision dimension in addition to network and accelerator co-explorations. Specifically, TRIPS with searched precision achieves a +5.96% higher accuracy and 4.4× FPS on ImageNet over (Jiang et al., 2020b).

Co-search on ASIC. We benchmark with NA-SAIC, the first exploration towards network / accelerator co-search targeting ASIC accelerators, with both the co-search results and their reported sequential optimization/hardware aware optimization results (Yang et al., 2020) on CIFAR-10 in Tab. 3. We can observe that compared with both co-search, sequential optimization, and hardware-aware optimization methods for

Table 4: Benchmark TRIPS over NHAS (Lin et al.) and DANCE (Choi et al., 2020) under the same precision setting.

| Co-search Methods | Accuracy (%) | Latency (ms) | Area (mm²) |
|---|---|---|---|
| NHAS (Lin et al.) | 70.74 | 1.58 | 5.87 |
| **TRIPS** | **71.70** | **1.25** | **5.50** |
| DANCE (Choi et al., 2020) | 68.70 | 8.13 | 2.73 |
| **TRIPS** | **72.20** | **2.85** | **2.12** |

exploring the ASIC design space, our TRIPS framework consistently achieves notably improved trade-offs between accuracy and energy delay product (EDP), which is energy multiplied with latency. In particular, we achieve a +0.17% ~ +1.81% higher accuracy with a 371.56 ~ 756.88× reduction in EDP. In the baseline implementations, most of the area is occupied by the support for heterogeneous functionalities, which leads to severe under utilization when executing one task, thus contributing to the surprisingly higher area and energy consumption.

We further benchmark TRIPS over other two co-search baselines targeting ASIC accelerators, i.e., NHAS (Lin et al.) and DANCE (Choi et al., 2020). In particular, we fix the precision of TRIPS to be 4-bit and 16-bit to fairly compare with (1) NHAS which adopts 4-bit and (2) DANCE which adopts 16-bit, respectively. As shown in Tab. 4, TRIPS achieves a 0.96%/3.5% higher accuracy and a 20.9%/64.9% reduction in latency together with a 6.3%/22.3% reduction in area consumption, as compared with NHAS and DANCE, respectively, verifying the superiority of our TRIPS.

### 4.4 ABLATION STUDIES ABOUT TRIPS

**Scalability under the same DSP.** In Fig. 5, we show the pareto frontier of our TRIPS framework under the same DSP constraint with different accuracy and FPS trade-offs on CIFAR-100 to show our TRIPS can handle and is scalable with a large range of DNN solutions.

**Effectiveness of heterogeneous sampling.** In addition to the example and analysis in Sec. 3.2, we further benchmark with the baseline that adopts the same sampling strategy for updating both the weights and precision. We integrate the baseline's sampling strategy into our TRIPS framework ($K = 2$ for all the experiments), termed as TRIPS w/o h-sampling, and show

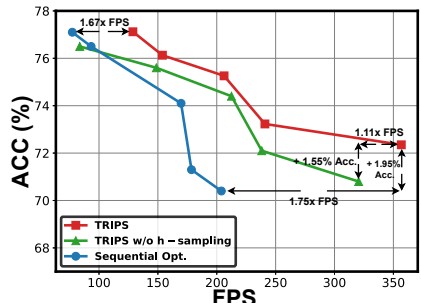

Figure 5: Accuracy vs. FPS trade-off of TRIPS, TRIPS w/o heterogeneous sampling, and the sequential optimization baseline on CIFAR-100.

the trade-offs between the achieved accuracy and FPS in Fig. 5. We find it tends to select lower precision choices which are harmful to the overall accuracy, which is consistently inferior than that of TRIPS with heterogeneous sampling, due to the inaccurate estimation for different precision ranking .

**Comparison with sequential optimization.** Due to the great flexibility on both software and hardware side, a natural baseline is to search the network and precision based on theoretical efficiency metrics (e.g., bit operations) and then search for the best matched accelerator given the searched network and precision from the first search. We benchmark over results from such a design flow in Fig. 5 on CIFAR-100 and observe that TRIPS consistently outperforms the sequential optimization baseline, e.g., a 1.95% higher accuracy with 1.75× FPS, indicating the poor correlation between theoretical efficiency and real-device efficiency measurement.

**More ablation studies about the accelerator search engine and visualization of the searched network, precision and accelerator can be found in Appendix C and D, respectively.**

### 5 CONCLUSION

We propose a Triple-Search (TRIPS) framework to jointly search for the network structure, precision, and accelerator in a differentiable manner. Our TRIPS framework adopts a heterogeneous sampling strategy and a novel co-search pipeline that integrates a generic differentiable accelerator search engine to achieve unbiased search with constant memory consumption. Extensive experiments validate the superiority of TRIPS over SOTA designs in terms of accuracy and efficiency.

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

## A    Training and Search Setting

**Search settings.** For searching on CIFAR-10/100 dataset, we use half of the dataset for updating supernet weight $\omega$ and the other half for updating network and precision parameter $\alpha$ and $\beta$. We search for 90 epochs with an initial gumbel softmax temperature 5 decayed by a factor 0.975 every epoch. For searching on ImageNet, we randomly sample 100 classes as a proxy search dataset and

use 80% data for updating $\omega$ and the other 20% for updating $\alpha$ and $\beta$. We pretrain the supernet by 30 epochs without updating network architecture and precision, then search for 90 epochs with an initial temperature 5 decayed by 0.956 every epoch, following (Wu et al., 2019). For both CIFAR-10/100 and ImageNet, we use initial learning rate 0.1 and an annealing cosine learning rate.

**Training settings.** For CIFAR-10/100, we train the derived network for 600 epochs with a 0.1 initial learning rate and an annealing cosine learning rate on a single NVIDIA RTX-2080Ti gpu following (Liu et al., 2018). For ImageNet, we adopt a 0.05 initial learning rate and an annealing cosine learning rate for 150 epochs with four NVIDIA Tesla V100 gpus.

## B  ACCELERATOR SEARCH SPACE

To flexibly maintain the balance between hardware resource consumption and throughput across different generated networks, we employ the chunk-wise pipeline styled accelerator architecture inspired by (Shen et al., 2017; Zhang et al., 2018b). To enable the automatic hardware optimization and explore as much as the performance frontier, we further free up the hardware configurations during the co-optimization. Adopted from (Chen et al., 2017; Zhang et al., 2015; Yang et al., 2016), these configurations, as illustrated in Fig. 1, cover **1)** parallel processing elements (PE) settings: number and inter-connections of PEs, **2)** buffer management: allocation of lower memory levels between input, weight and output, **3)** tiling and scheduling of MAC(Multiply and Accumulate) computations and **4)** layer allocation: ways to assign each layer to the corresponding pipeline stage (sub-accelerator). All the above configurations are formatted and maintained through vectors of options to be compatible with the optimization formulation in Sec. 3. Taking AlexNet as an example workload, the total accelerator space size can reach up to $10^5$ for each sub-accelerator and the space can go exponentially larger as the number of sub-accelerator (pipeline chunks) increases.

## C  ABLATION STUDIES ABOUT THE ACCELERATOR SEARCH ENGINE

The proposed accelerator search engine is one key enabler of our TRIPS framework. To evaluate its efficacy, we compare the accelerator efficiency of the TRIPS generated accelerators with SOTA accelerators under the same datasets and models. For FPGA-based accelerators, we consider three representative including (Qiu et al., 2016; Xiao et al., 2017; Zhang et al., 2018b) on two DNN models (AlexNet and VGG16) on ImageNet. For a fair comparison, when using our own engine to generate optimal accelerators, we adopt the same precision and FPGA resource as the baselines. The results in Tab. 5 show that the TRIPS generated accelerators outperform **both SOTA expert-designed and tool-generated** accelerators under the same dataset, DNNs, and FPGA resources. For example, the TRIPS generated accelerators achieve up to 2.16× increase in throughput on VGG16. The consistent better performance of our auto generated accelerators validates the effectiveness of our accelerator search engine in navigating the large and discrete design space of DNN accelerators to search for optimal DNN accelerators.

Table 5: TRIPS generated FPGA accelerators vs. SOTA FPGA accelerators built on top of Zynq XC70Z45 with 200 Mhz applied on different networks with fixed 16-bit on ImageNet.

| | (Zhang et al., 2018b) | (Xiao et al., 2017) | (Qiu et al., 2016) | **TRIPS generated** | (Zhang et al., 2018b) | **TRIPS generated** |
|---|---|---|---|---|---|---|
| Network | VGG16 | VGG16 | VGG16 | VGG16 | AlexNet | AlexNet |
| Resource Utilization | 680/900 DSP | 824/900 DSP | 780/900 DSP | 723/900 DSP | 808/900 DSP | 704/900 DSP |
| Performance (GOP/s) | 262 | 230 | 137 | **291** | 247 | **272** |

## D  VISUALIZATION OF SEARCHED NETWORK, PRECISION, AND ACCELERATOR.

Fig. 6 visualizes the searched network, precision, and accelerator achieved a 72.2% top-1 accuracy on ImageNet and 110 FPS on ZC706 FPGA. We conclude the insights below.

**Insights for the searched network.** We find that wide-shallow networks will better favor real device efficiency on ZC706 FPGA while achieve a similar accuracy. We conjecture the reason is that wider networks offer more opportunities for feature/channel-wise parallelism when batch size equals to one, leading to higher resource utilization and thus overall higher throughput.

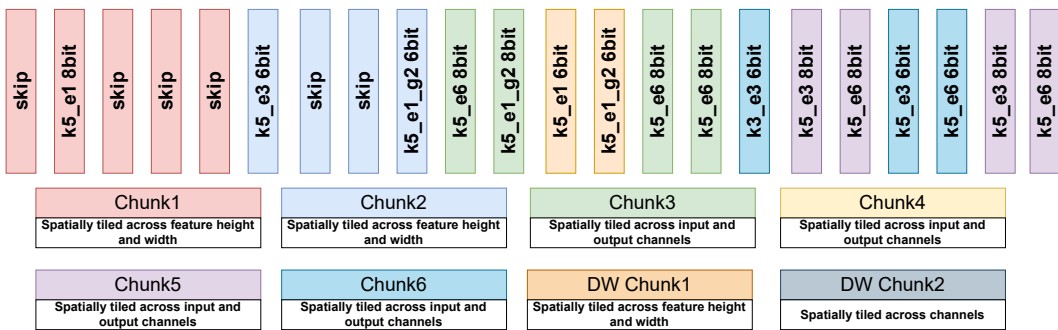

Figure 6: Visualization of the searched network, precision, and accelerator that achieves a 72.2% top -1 accuracy on ImageNet and 110 FPS on ZC706 FPGA. The block definition follows (Wu et al., 2019).

**Insights for the searched accelerator of TRIPS.** The whole network is partitioned into multiple pipeline chunks to prioritize the resulting throughput, with each color representing one chunk. Deterministically, two blocks with different quantization schemes will not be put into one chunk due to hardware constraints. Adopted from the design in (Shen et al., 2017), there is no dependency of computation among blocks, so the pipeline chunks can take non-consecutive layers and better group the layers with similar dimensions together. Generally, as observed in Fig. 6 the chunks which take mostly the early blocks of the network favor spatially tiling the feature map height and width as it offers more parallelism, while the later chunks tends to choose the architecture to tile channels as, after down-sampling, parallelism opportunity is more prominent along channel dimensions.

