# OpenReview forum: "Triple-Search: Differentiable Joint-Search of Networks, Precision, and Accelerators"
_ICLR.cc/2021/Conference — Reject_

### Official Review · AnonReviewer4 · 2020-10-26
**Review for Submission 2589**

**Rating:** 6
**Confidence:** 2

**Review:**

##########################################################################
Summary:

The authors present TRIPS, a Triple-Search framework to jointly search for the optimal network, precision and accelerator for a given task with max accuracy and efficiency in a differentiable manner. TRIPS focuses on efficiently exploring the large search space that previous reinforcement learning based solution cannot afford to due to poor scalability.

The authors propose a heterogeneous sampling approach that enables simultaneous update of weights and precision without biasing the precisions options or exploding the memory consumption. Further, they also formulate a novel co-search pipeline that enables a differentiable search for optimal accelerator.

Lastly, the authors present ablation studies and comparison with SOTA solutions that either solve the triple search problem described above or search (network, precision, accelerator) over a subset of its search space.

##########################################################################
Reasons for score:

The paper presents a thorough description of TRIPS a joint search algorithm that enables scalable search of optimal network, precision and accelerator for maximum accuracy and efficiency. The experiments demonstrate the improvement in accuracy and efficiency compared to a wide range of previous solutions. Further ablation studies show breakdowns of accuracy and efficiency improvements derived from key enablers of TRIPS. With a few clarifications for the questions mentioned in the Cons section this paper should be accepted.


##########################################################################
Pros:

1.	This work presents a novel attempt to explore the large search space of network, precision and accelerator design, including detailed parameters such as tiling strategies, buffer sizing, processing elements (PE) count and connections etc.
2.	Section 2 presents a detailed overview of the past papers that have attempted to optimize all subsets of the search space explored by the proposed solutions.
3.	Authors do a great job of either estimating or collecting data for comparing the proposed solution with the many different approaches discussed in Section 2.
4.	Figure 3 demonstrates good data establishing the superiority of TRIPS optimal solution compared with other baselines. It would be helpful if the authors could shed light on how they tune TRIPS to generate a range of different solutions that tradeoff max accuracy with max FPS.
5.	Figure 4 presents a useful comparison to previous solutions that do not optimize over precision space. TRIPS shows better accuracy and FPS even with fixed precision.
6.	Ablation studies in section 4.3, are useful in breaking down the incremental improvement in the accuracy vs FPS tradeoff curve of TRIPS.
7.	Appendix sections provide useful details about TRIPS training and accelerator design space explored. Additionally, section C shows another useful contrast of TRIPS with previous solutions based on SOTA expert-designed and tool-generated solutions. Remaining sections provide useful insights on the searched space.

##########################################################################
Cons:

1.	Authors should consider citing the following in Section 2, DNN accelerators para, since it explores mapping network to accelerator theme of this section:
Gao, Mingyu, et al. "Tetris: Scalable and efficient neural network acceleration with 3d memory." Proceedings of the Twenty-Second International Conference on Architectural Support for Programming Languages and Operating Systems. 2017.
2.	Towards the end of Section 2, authors mention that TRIPS selects network, precision and accelerator that enable better transferability to other tasks/applications. However, there are no results or discussion on transferability of TRIPS models to other applications, consider removing this line.
3.	While Section 3 is comprehensive in its detailed description of TRIPS implementation and key enablers. It is somewhat difficult to read, consider reordering the implementation discussion and Section 3.2/3.3.
4.	Section 4.2, it’s not clear why the authors selected the hardware limitation of 512DSP units.
5.	Further, Section 4.2, Co-exploration of networks and accelerators part, compares previous solutions that optimize network and accelerator while keeping the precision fixed (searched optimal value). It would be helpful if the authors added the precisions used for the previous work datapoints as well. Are the selected datapoints for previous papers, the max accuracy points? How do authors select the 16bit value they choose for Figure 4 fixed precision TRIPS?
6.	Additionally, the authors do not present TRIPS-16bit datapoint for Figure 4c), please add that.
7.	The analysis presented in Table 2 seems highly skewed, with almost 1000x improvement in Area and EDP. Knowing that the support for heterogenous functionalities is costing baseline ASIC implementations dearly makes the comparison unfair. I recommend that authors attempt to minimize the impact of such features, since they make the baseline ASIC accelerators more general which the proposed solution does not support.

##########################################################################
Questions during rebuttal period:

Kindly address a few questions in the Cons section.

#########################################################################
Some typos:
1.	Section 1: “to optimize the acceleration efficiency for* a* given* DNN”
2.	Section 2: Hardware-aware NAS subsection, “acceleration efficiency, thus can lead* to sub-optimal solutions.”
3.	Section 2: Co-exploration/search techniques subsection: “Built upon prior art*, our”
4.	Section 4.3: Comparison with sequential optimization subsection: “and hardware side, a natural* design”
5.	Appendix D: Insights for the searched network subsection:  “find that while wide-shallow* networks”, reword the whole sentence.

---

> ### Author Response · Authors · 2020-11-23
> **Response to Reviewer 4**
>
> Thank you very much for recognizing our contributions and for your valuable suggestions. Below are our answers to these questions:
>
> **1. Missing references and typos.**
>
> Thanks a lot for pointing out these. We will revise them and more carefully proofread in the final version.
>
> **2. Inaccurate claims.**
>
> Thanks for accurately pointing out this. We will remove the claim and leave transferability of TRIPS to other applications as one of our future works.
>
> **3. Organization of Section 3.**
>
> Good suggestion, thank you! We will reorganize this section accordingly and improve writing clarity in the final version.
>
> **4. Reasons for using 512 DSP limits.**
>
> We conducted all experiments under the same DSP limits for a fair comparison, and adopted the 512 DSP limit following the baseline “Best of Both Worlds: AutoML Codesign of a CNN and its Hardware Accelerator” (M. Abdelfattah, DAC’20) after confirming with the authors.
>
> **5. Details about the benchmark in Sec. 4.2.**
>
> Thanks for your suggestion. We will further clarify the precision for the baselines in the final version. Specifically, we select the datapoint of the best reported accuracy from the baselines’ papers. In Figure 4, all the baselines are in 16-bit, so we provide TRIPS with 16-bit for a fair comparison.
>
> **6. TRIPS with fixed 16-bit in Figure 4c.**
>
> Thanks a lot for catching this. Following your suggestion, we conducted the experiment and find that TRIPS with fixed 16-bit achieves 75.7% accuracy and an FPS of 31.0 on ImageNet, i.e. a 5.46% higher accuracy with 2.9x better FPS compared with the baseline in Figure 4c, under the same DSP constraint. We will add this in the final version.
>
> **7. Fair comparisons on ASIC.**
>
> Thank you for your insightful comment. We benchmarked with NASAIC since they are the first exploration towards network / accelerator co-search targeting ASIC accelerators. Since they are not open-sourced, it’s difficult for us to fairly compare with them when their heterogeneous functionalities are turned off. Following your suggestion and to fairly benchmark TRIPS when searching for ASIC accelerators, we have conducted additional experiments to benchmark with another two co-search baselines: “Neural-Hardware Architecture Search” (Y.Lin, NeurIPS Workshop, 2019) and “Dance: Differentiable Accelerator/Network Co-Exploration” (K. Choi, arXiv’20). In particular, we fix the precision of TRIPS to be 4-bit and 16-bit to fairly compare with (1) NHAS which adopts 4-bit and (2) DANCE which adopts 16-bit, respectively.  As shown in the table below, TRIPS achieves a 0.96%/3.5% higher accuracy and a 20.9%/64.9% reduction in latency, together with a 6.3%/22.3% reduction in area consumption, as compared with NHAS and DANCE, respectively, verifying the superiority of our TRIPS. We will clarify the skewed aspect of comparing with NASAIC and add the above experiment results in our final version.
>
> | Method    | Accuracy (%) | Latency (ms) | Area (mm^2) |
> |-----------|--------------|--------------|-------------|
> |    NHAS   |     70.4     |     1.58     |     5.87    |
> | **TRIPS** |   **71.7**   |   **1.25**   |   **5.50**  |
> |    DANCE  |     68.7     |     8.13     |     2.73    |
> | **TRIPS** |   **72.2**   |   **2.85**   |   **2.12**  |

---

### Official Review · AnonReviewer3 · 2020-10-28

**Rating:** 5
**Confidence:** 5

**Review:**

Review:
This paper proposes Triple-Search (TRIPS), a differentiable framework of jointly searching for network architecture, quantization precision, and accelerator parameters. To address the dilemma between exploding training memory and biased search, the proposed framework leverages heterogeneous sampling where soft Gumbel Softmax is used for weight update and hard Gumbel Softmax is used for probabilities \beta. To integrate accelerator search, hard Gumbel Softmax is used on hardware design choices and the overall hardware cost is used for penalization. Experiments are conducted on the FPGA platform for CIFAR and ImageNet dataset to show the superiority of TRIPS over NAS-only methods.

Pros:

+ The idea of co-searching hardware architecture, neural architecture, and quantization precision is promising.

+ It's nice to see the ablation study on sequential optimization and joint optimization.


Concerns:

- The paper is not well organized. Though TRIPS is a co-search framework, the paper mostly elaborates on the quantization searching. The modeling of hardware and the design space of the accelerator search are not well introduced.

- The challenge of exploding training memory comes from the differentiable searching. However, the authors did not explain why the differentiable search is a must. The state-of-the-art NAS algorithms tend to first train a shared weight hyper net and then perform searching algorithms such as the evolutionary algorithm. In this case, the main challenge in this paper does not exist.

- The comparison in Figure 3 may not be fair. The available hardware resources of the FPGA and mobile phones are different. It would be better if TRPIS results can be compared on the same hardware platform.

- The following papers also search quantization precision during NAS. It would better if the paper can show the comparison between TRIPS and the following methods.

[1] Yujun Lin, Driss Hafdi, Kuan Wang, Zhijian Liu, and Song Han. Neural-Hardware Architecture Search. NeurIPS Workshop, 2019.

[2] Dimitrios Stamoulis, Ruizhou Ding, Di Wang, Dimitrios Lymberopoulos, Bodhi Priyantha, Jie Liu, and Diana Marculescu. Single-path nas: Designing hardware-efficient convnets in less than 4 hours. In Joint European Conference on Machine Learning and Knowledge Discovery in Databases, pp. 481–497. Springer, 2019.

- For CIFAR experiments in Figure 4, it would be better to show the improvement over fixed 8bit and fixed 4bit instead of 16bit.

- For ASIC experiments, what is the accelerator design space? The results in Table 2 may not be fair. NASAIC uses MAESTRO as the estimation tool while TRIPS uses Timeloop as the estimation tool. If both NASAIC and TRIPS only search hardware parameters, does NASAIC and TRIPS search on the same hardware architecture (such as Eyeriss/NVDLA)?

---

> ### Author Response · Authors · 2020-11-23
> **Response to Reviewer 3: Part 1**
>
> Thank you very much for your valuable suggestions. Below are our detailed response to your questions/concerns:
>
> (Note that here's the Part 1 of our responses, see Part 2 for more responses.)
>
> **1. More details about hardware search.**
>
> As Reviewer 4 also pointed out “Appendix sections provide useful details about TRIPS training and accelerator design space explored”. We will add more details about the hardware template in the final version. Here’s an overview of the design space of the accelerator search for the proposed hardware search engine in Sec. 3.3.
> As introduced in Sec. 4.1, we consider both FPGA and ASIC accelerators to evaluate TRIPS with detailed design knobs discussed in the appendix B. In particular, we adopt a unified template for both the FPGA and ASIC accelerators, which is a parameterized chunk-based pipeline micro-architecture inspired by “Maximizing CNN Accelerator Efficiency Through Resource Partitioning” (Y. Shen, ISCA’17).
>
> Implementation: The hardware/micro-architecture template comprises multiple sub-accelerators (i.e., chunks) and executes DNNs in a pipeline fashion. In particular, each chunk is assigned with multiple but not necessarily consecutive layers which are executed sequentially within the chunk. Similar to Eyeriss, each chunk consists of levels of buffers/memories (e.g., on-chip buffer and local register files) and processing elements (PEs) to facilitate data reuses and parallelism with searchable design knobs such as PE interconnections (i.e., Network-on-chip), allocated buffer sizes, MAC operations’ scheduling and tiling (i.e., dataflows), and so on (see more details in the appendix B).
>
> **2. Motivation for using differentiable NAS.**
>
> Thanks again for your comment.  We will clarify more about this in the final version.
>
> First, as we emphasized in our manuscript’s introduction, one major challenge for co-searching for the network, precision, and accelerator is the huge joint search space, which can lead to an unacceptable search time if using previous RL-based co-search methods. Furthermore, we pointed out that differentiable NAS is one of the most accessible NAS methods to the community thanks to its low search cost, which can be further extended by our proposed search strategy (described in Sec. 3.1 and Sec. 3.2) to largely improve the search efficiency, i.e., our TRIPS search strategy makes it possible to consider a much larger joint search space and enables more practical joint search solutions, as compared with existing co-search works.
>
> Second, we agree that the two one-shot NAS methods you mentioned can be potential solutions for our target triple search, but they still require a large amount of time for supernet pretraining even without considering the precision search, as verified in our experiments. In the table below, we compare the search space size and search time  of TRIPS with both RL-based and one-shot NAS methods using the reported data from the baselines’ original papers, where (1) [1][2] are RL-based network/accelerator co-search methods and [3][4][5] are one-shot NAS methods (including the two you mentioned) and (2) [4] is another pioneering work which extends [3] to include precision search. We can see that TRIPS consistently require notably less search time under the largest joint search space on all the tasks. In particular, compared with the one-shot NAS methods, TRIPS achieves 3.6x ~ 30x less search time on ImageNet, while being end-to-end. Furthermore, experiments in Sec. 4.2 of our manuscript show that TRIPS achieves the best performance among [1][2][4].
>
> | Method | Dataset | Network Space | Accelerator Space | Precision Space | Joint Space | Search Time (GPU hours) |
> |-|-|-|-|-|-|-|
> | HS-Co-Opt [1] | CIFAR-10 | 1.15E+18 | - | - | 1.15E+18 | 103.9 |
> | TRIPS | CIFAR-10 | 9.85E+20 | 2.24E+27 | 2.40E+15 | **5.30E+63** | **6** |
> | BSW [2] | CIFAR-100 | 4.20E+05 | 8.64E+03 | - | 3.63E+09 | 5184 |
> | TRIPS | CIFAR-100 | 9.85E+20 | 2.24E+27 | 2.40E+15 | **5.30E+63** | **12** |
> | HS-Co-Opt [1] | ImageNet | 2.22E+18 | - | - | 2.22E+18 | 266.8 |
> | Once-For-All[3] | ImageNet | 2.00E+19 | - | - | 2.00E+19 | 1200 |
> | APQ [4] | ImageNet | 1.00E+35 | - | 1.00E+10 | 1.00E+45 | 2400 |
> | Single [5] | ImageNet | 7.00E+21 | - | - | 7.00E+21 | 288 |
> | TRIPS | ImageNet | 9.85E+20 | 2.24E+27 | 2.40E+15 | **5.30E+63** | **80** |
>
>
> [1]: “Hardware/Software Co-Exploration of Neural Architectures.” (W. Jiang, TCAD’20)
>
> [2]: “Best of Both Worlds: AutoML Codesign of a CNN and its Hardware Accelerator.” (M. Abdelfattah, DAC’20)
>
> [3]: “Once-For-All: Train One Network and Specialize It for Efficient Deployment.” (H. Cai, ICLR’20)
>
> [4]: “APQ: Joint Search for Network Architecture, Pruning and Quantization Policy.” (T. Wang, CVPR’20)
>
> [5]: “Single One-Shot Neural Architecture Search With Uniform Sampling.” (Z. Guo, ECCV’20)

---

> ### Author Response · Authors · 2020-11-23
> **Response to Reviewer 3: Part 2**
>
> (Note that here's the Part 2 of our responses, see Part 1 for more responses.)
>
> **3. Comparison in Fig. 3.**
>
> As we described in Sec. 4.2, we indeed compare all the baselines in Fig. 3 on the same platform (FPGA) under the same DSP constraint (512). In particular, we search for the optimal accelerator design based on our accelerator space for APQ, MP-NAS, and SOTA hardware-aware NAS methods to ensure a fair comparison. In this way, we make sure the comparison is based on the same hardware platform with the same constraint, the same accelerator search algorithm, and the same accelerator search space.
>
> **4. Benchmark with NHAS and Single-path NAS.**
>
> **Benchmark with NHAS:** We have added the benchmark with NHAS below. For a fair comparison with NHAS which adopts a uniform 4-bit precision, we fix the precision to be 4-bit and co-search the network and accelerator based on our ASIC template, while enhancing a constraint of having a comparable area as NHAS. As shown in the table below, TRIPS achieves a 0.96% better accuracy together with a 20.89% reduction in latency and 6.3% improvement in area consumption, as compared with NHAS. We will cite and benchmark with NHAS in the final version.
>
> | Method    | Accuracy (%) | Latency (ms) | Area (mm^2) |
> |-----------|--------------|--------------|-------------|
> |    NHAS   |     70.4     |     1.58     |     5.87    |
> | **TRIPS** |   **71.7**   |   **1.25**   |     **5.50**   |
>
> **Benchmark with Single-path NAS:** The single-path NAS work you mentioned doesn’t conduct precision search experiments and we conjecture the correct paper you meant is “Single One-Shot Neural Architecture Search With Uniform Sampling” (Z. Guo, ECCV’20) which considers precision search. Nonetheless, we have conducted experiments to benchmark with both of them as shown in the table below. We can see that TRIPS outperforms both baselines with a 1.24% / 2.3% higher accuracy and a 1.5x / 6.6x higher FPS, respectively. In particular, the single one-shot NAS baseline with searched precisions and channels assumes a fixed ResNet-34 backbone, which is one reason resulting in its inefficiency.
>
> | Method    | Accuracy (%) |    FPS       |
> |-----------|--------------|--------------|
> |Single-path NAS (16-bit) |   74.96    |   30.74     |
> |Single One-Shot (ResNet-34 with searched precisions and channels)         |   73.90     |   7.06      |
> | **TRIPS** |   **76.20**   |   **46.27**   |
>
> **5. Precision choices for Fig. 4.**
>
> We agree with your suggestion and would love to do that. However, the baselines in Fig. 4 are not open-sourced, so we directly pick their best reported results (16-bit only) from their original papers, and report the results of TRIPS with both fixed 16-bit and searched precision for a fair comparison. More benchmarks with mixed-precision baselines can be found in Fig. 3.
>
> **6. Details about the accelerator search.**
>
> As mentioned in our first response item to you, we adopt a chunk-based pipeline accelerator/micro-architecture inspired by “Maximizing CNN Accelerator Efficiency Through Resource Partitioning” (Y. Shen, ISCA’17) for the ASIC experiments with the detailed design knobs discussed in the appendix B. Therefore, TRIPS and NASAIC do not just search on the hardware architecture, and TRIPS’s search space is much larger and flexible (2e27 vs. 1e6) than that of NASAIC for better unleashing the great potential of co-searching. Furthermore, our large joint search space can be hardly handled by the RL-based NASAIC due to the prohibitive search cost of RL-based methods, which is effectively tackled by the proposed search strategy in TRIPS (see Sec. 3).
>
> For the estimation tool,  both Timeloop (A. Parashar, ISPASS’19) and MAESTRO (H. Kwon, MICRO’19) are capable of accurately estimating the hardware cost with similar estimation results. As shown in Fig. 9 of MAESTRO and Fig. 10 of Timeloop, their estimated results are both close ( within 10%)  to the actual reported ones from Eyeriss. Furthermore, as claimed in Sec. 4.3 of MAESTRO, its energy model can be replaced by Accelergy (Y. Wu, ICCAD’19), which is also the energy model of Timeloop in our implementation, indicating the interchangeability of the two estimation tools.

---

### Official Review · AnonReviewer1 · 2020-10-28
**Official Blind Review #1**

**Rating:** 5
**Confidence:** 4

**Review:**

This paper introduces Triple-Search for jointly optimizing neural network architecture, quantization policy, and hardware architecture. Specifically, it proposes a heterogeneous sampling strategy to tackle the dilemma between exploding memory cost and biased search. Besides, it also integrates a differentiable hardware search engine to support co-search in a differentiable manner.

Pros:
1. Co-searching neural network architecture, quantization policy, and hardware architecture is practically important.
2. Experiment results show clear improvements over the baselines.

Cons:
1. I think this paper is not well motivated. I am not convinced that the challenge (dilemma between exploding memory cost and biased search) that this paper aims to address is a critical problem in practice. This challenge exists because the authors follow the differentiable NAS framework. However, for the chain-like design space studied in this paper, recent state-of-the-art NAS techniques [1,2] have shown that weight training can be decoupled from architecture search, and differentiable search is not a must in practice. In this paper, I do not find any discussions to support why we must need a differentiable AutoML framework for the co-searching task.
2. The hardware part of the proposed framework is too short and not clearly explained. Please elaborate more on this part.
3. The same co-searching problem has been studied in [3]. I think the authors should compare the proposed method with [3].

Overall, I think co-searching neural network architecture, quantization policy, and hardware architecture is an important task. The experiment results also look good. However, I find the motivation of having a differentiable framework is not well supported, the hardware part of the proposed framework is not clearly explained, and some closely related papers are missing. Therefore, I recommend "Marginally below acceptance threshold".

[1] Once for all: Train one network and specialize it for efficient deployment. ICLR 2020.

[2] Single path one-shot neural architecture search with uniform sampling. arXiv preprint arXiv:1904.00420 (2019).

[3] Neural-Hardware Architecture Search. NeurIPS 2019 Workshop on Machine Learning for Systems.

---

> ### Author Response · Authors · 2020-11-23
> **Response to Reviewer 1: Part 1**
>
> Thank you very much for your valuable suggestions.  Below are our detailed response to your questions/concerns:
>
> (Note that here's the Part 1 of our responses, see Part 2 for more responses.)
>
> **1. Motivation for using differentiable NAS.**
>
> Thanks again for your comment.  We will clarify more about this in the final version.
>
> First, as we emphasized in our manuscript’s introduction, one major challenge for co-searching for the network, precision, and accelerator is the huge joint search space, which can lead to an unacceptable search time if using previous RL-based co-search methods. Furthermore, we pointed out that differentiable NAS is one of the most accessible NAS methods to the community thanks to its low search cost, which can be further extended by our proposed search strategy (described in Sec. 3.1 and Sec. 3.2) to largely improve the search efficiency, i.e., our TRIPS search strategy makes it possible to consider a much larger joint search space and enables more practical joint search solutions, as compared with existing co-search works.
>
> Second, we agree that the two one-shot NAS methods you mentioned can be potential solutions for our target triple search, but they still require a large amount of time for supernet pretraining even without considering the precision search. As shown in the table below, we compare the search space size and search time  of TRIPS with both RL-based and one-shot NAS methods using the reported data from the baselines’ original papers, where (1) [1][2] are RL-based network/accelerator co-search methods and [3][4][5] are one-shot NAS methods (including the two you mentioned) and (2) [4] is another pioneering work which extends [3] to include precision search. We can see that TRIPS consistently require notably less search time under the largest joint search space on all the tasks. In particular, compared with the one-shot NAS methods, TRIPS achieves 3.6x ~ 30x less search time on ImageNet, while being end-to-end. Furthermore, experiments in Sec. 4.2 of our manuscript show that TRIPS achieves the best performance among [1][2][4].
>
>
> |      Method     |  Dataset  | Network Space | Accelerator Space | Precision Space | Joint Space | Search Time (GPU hours) |
> |:---------------:|:---------:|:-------------:|:-----------------:|:---------------:|:-----------:|:-----------------------:|
> |  HS-Co-Opt [1]  |  CIFAR-10 |    1.15E+18   |         -        |        -       |   1.15E+18  |          103.9          |
> |    **TRIPS**    |  CIFAR-10 |    9.85E+20   |      2.24E+27     |     2.40E+15    |   **5.30E+63**  |          **6**          |
> |     BSW [2]     | CIFAR-100 |    4.20E+05   |      8.64E+03     |        -       |   3.63E+09  |           5184          |
> |    **TRIPS**    | CIFAR-100 |    9.85E+20   |      2.24E+27     |     2.40E+15    |   **5.30E+63**  |          **12**         |
> |  HS-Co-Opt [1]  |  ImageNet |    2.22E+18   |         -         |        -        |   2.22E+18  |          266.8          |
> | Once-For-All[3] |  ImageNet |    2.00E+19    |         -         |        -       |   2.00E+19   |           1200          |
> |     APQ [4]     |  ImageNet |    1.00E+35   |         -         |     1.00E+10    |   1.00E+45  |           2400          |
> |    One-shot [5]   |  ImageNet |    7.00E+21   |         -         |        -        |   7.00E+21  |           288           |
> |    **TRIPS**    |  ImageNet |    9.85E+20   |      2.24E+27     |     2.40E+15    |   **5.30E+63**  |          **80**         |
>
> [1]: “Hardware/Software Co-Exploration of Neural Architectures.” (W. Jiang, TCAD’20)
>
> [2]: “Best of Both Worlds: AutoML Codesign of a CNN and its Hardware Accelerator.” (M. Abdelfattah, DAC’20)
>
> [3]: “Once-For-All: Train One Network and Specialize It for Efficient Deployment.” (H. Cai, ICLR’20)
>
> [4]: “APQ: Joint Search for Network Architecture, Pruning and Quantization Policy.” (T. Wang, CVPR’20)
>
> [5]: “Single One-Shot Neural Architecture Search With Uniform Sampling.” (Z. Guo, ECCV’20)

---

> ### Author Response · Authors · 2020-11-23
> **Response to Reviewer 1: Part 2**
>
> (Note that here's the Part 2 of our responses, see Part 1 for more responses.)
>
> **2. More details about hardware search.**
>
> As Reviewer 4 also pointed out “Appendix sections provide useful details about TRIPS training and accelerator design space explored”. We will add more details about the hardware template in the final version.
>
> As introduced in Sec. 4.1, we consider both FPGA and ASIC accelerators to evaluate TRIPS with detailed design knobs discussed in the appendix B of our manuscript. In particular, we adopt a unified template for both the FPGA and ASIC accelerators, which is a parameterized chunk-based pipeline micro-architecture inspired by “Maximizing CNN Accelerator Efficiency Through Resource Partitioning” (Y. Shen, ISCA’17).
>
> Implementation: The hardware/micro-architecture template comprises multiple sub-accelerators (i.e., chunks) and executes DNNs in a pipeline fashion. In particular, each chunk is assigned with multiple but not necessarily consecutive layers which are executed sequentially within the chunk. Similar to Eyeriss, each chunk consists of levels of buffers/memories (e.g., on-chip buffer and local register files) and processing elements (PEs) to facilitate data reuses and parallelism with searchable design knobs such as PE interconnections (i.e., Network-on-chip), allocated buffer sizes, MAC operations’ scheduling and tiling (i.e., dataflow), and so on (see more details in the appendix B).
>
> **3. Benchmark with NHAS (Y. Lin, NeurIPSW’19).**
>
> We have added the benchmark with NHAS below. For a fair comparison with NHAS which adopts a uniform 4-bit precision, we fix the precision to be 4-bit and co-search the network and accelerator based on our ASIC template, while enhancing a constraint of having a comparable area as NHAS. As shown in the table below, TRIPS achieves a 0.96% better accuracy together with a 20.89% reduction in latency and 6.3% improvement in area consumption, as compared with NHAS. We will cite and benchmark with NHAS in the final version.
>
> | Method    | Accuracy (%) | Latency (ms) | Area (mm^2) |
> |-----------|--------------|--------------|-------------|
> |    NHAS   |     70.4     |     1.58     |     5.87    |
> | **TRIPS** |   **71.7**   |   **1.25**   |     **5.50**    |
>
> **4. Missing references.**
>
> Thanks for pointing out and we will add all these references.

---

### Official Review · AnonReviewer2 · 2020-10-29
**Ambitious problem**

**Rating:** 6
**Confidence:** 3

**Review:**

Not an expert but an interested outsider trying to exercise the best judgement.

The goal is ambitious and is something that lots of people want to solve, so it's certainly worth a stab at. The nice thing about the paper is that the evaluation is pretty solid, with FPGA implementations and simulations for the ASIC flow (not that those simulators are perfect, but they are at least widely-used in the community so using them is probably OK.)

* Second paragraph on page 2, can you precisely define what you mean by "path" there?
* What exactly is your hardware cost model, L_{cost}?
* Could you give theoretical analysis as to why you optimization method will converge? It doesn't seem to be using a standard optimization method, which is perfectly fine, but then theoretically analyzing the proposed method would be important. Otherwise, the proposed method looks purely empirical, and is thus perhaps better suited for the CAD community.
* What basic assumptions about the hardware are you making? Is it a GPU-like SIMD/SIMT architecture, or TPU-like systolic array, or eyeriss-like thing that relies on pretty beefy interconnect? After all, you have a hardware template with several knobs and are searching the parameters of those knobs, so showing the hardware template would make the paper more clearer.

---

> ### Author Response · Authors · 2020-11-23
> **Response to Reviewer 2**
>
> Thank you very much for recognizing the importance of our target problem and for your valuable suggestions. Below are our answers to you questions:
>
> **1. Definition of path.**
>
> One path denotes one candidate operator between two activation nodes, and activated paths are the operators that participate in the forward or backward pass during search. We will clarify this in the final version.
>
> **2. Hardware cost model L_{cost}.**
>
> For FPGA, we use “latency” as the hardware cost loss since the target metric is FPS; and for ASIC, we use “EDP (Energy-Delay-Product)” as the hardware cost loss.
>
> **3. Convergence of the optimization method.**
>
> First, we would like to clarify that we in fact use a standard optimization method, i.e., bi-level optimization, which is a commonly used optimization method in state-of-the-art neural architecture search (NAS) works (e.g., DARTS (H. Liu, CVPR’19) and FBNet (B. Wu, CVPR’19)). The difference is that we use bi-level optimization to successfully achieve a triple search of matched bit-widths, neural networks, and accelerators to maximize both the network accuracy and the accelerator efficiency, handling a much larger joint search space as compared to standard NAS methods while maintaining an affordable search cost.  Specifically, we find that (1) the number of activated paths (i.e., operators) during backward propagation is critical in balancing the network accuracy and accelerator efficiency, and (2) an intermediate choice between the single-path NAS (e.g., GDAS (X. Dong, CVPR’19) and DSNAS (S. Hu, CVPR’20)) and the full-path NAS (e.g., DARTS and FBNet), both of which the convergences have been empirically verified in many NAS works, works the best in our triple search setting.
>
> Second, to further validate the convergence of searching with different numbers of activated paths under a fixed hardware cost constraint, we conduct an ablation study and summarize the final accuracy of the searched networks in the table below. We can observe that (1) TRIPS consistently converges well under all the cases, and (2) activating more paths generally leads to marginally better accuracies at the cost of larger GPU memory (see Fig. 2(a)) and search time.
>
> | #Active paths | 2     | 4     | 6     | 8     |
> |---------------|-------|-------|-------|-------|
> | Accuracy (%)  | 72.15 | 72.47 | 72.78 | 72.33 |
>
> We keep the theoretical analysis of our optimization method as one of our most exciting future works.
>
> **4. More details about the hardware template.**
>
> As Reviewer 4 also pointed out “Appendix sections provide useful details about TRIPS training and accelerator design space explored”. As introduced in Sec. 4.1, we consider both FPGA and ASIC accelerators to evaluate TRIPS with detailed design knobs discussed in the appendix B. In particular, we adopt a unified template for both the FPGA and ASIC accelerators, which is a parameterized chunk-based pipeline micro-architecture inspired by “Maximizing CNN Accelerator Efficiency Through Resource Partitioning” (Y. Shen, ISCA’17).
>
> Implementation: The hardware/micro-architecture template comprises multiple sub-accelerators (i.e., chunks) and executes DNNs in a pipeline fashion. In particular, each chunk is assigned with multiple but not necessarily consecutive layers which are executed sequentially within the chunk. Similar to Eyeriss, each chunk consists of levels of buffers/memories (e.g., on-chip buffer and local register files) and processing elements (PEs) to facilitate data reuses and parallelism with searchable design knobs such as PE interconnections (i.e., Network-on-chip), allocated buffer sizes, MAC operations’ scheduling and tiling (i.e., dataflows), and so on (see more details in the appendix B).
>
> We will add more details about the hardware template in the final version.

---

> > ### Comment · AnonReviewer2 · 2020-11-23
> > **Thanks for the responses. A few more comments/thoughts.**
> >
> > 1. With respect to your definition of path, seems like it's similar to quite a few prior work. See below for instance. Can you confirm whether they are the same as yours?
> > * https://ieeexplore.ieee.org/document/9251936
> > * https://openaccess.thecvf.com/content_cvpr_2018/papers/Wang_Interpret_Neural_Networks_CVPR_2018_paper.pdf
> > * https://openaccess.thecvf.com/content_CVPR_2019/papers/Qiu_Adversarial_Defense_Through_Network_Profiling_Based_Path_Extraction_CVPR_2019_paper.pdf
> >
> > 2. With respect to the hardware template, thanks for the clarification. Could you comment the general applicability of your search approach, especially if users are targeting a different hardware architecture? What you pick is fine I suppose, but it's arguable less mainstream than say TPU's systolic array. Most of the commercial DNN accelerators are perhaps just GEMM accelerators (correct me if I am wrong). How much more work a user would have to do if they were to target a different, perhaps more mainstream, accelerator?

---

> > > ### Author Response · Authors · 2020-11-23
> > > **Response to the the additional comments of Review 2**
> > >
> > > Thanks very much for your quick response and insightful questions. Below are our answers to the questions:
> > >
> > > **1. Clarifications about the definition of path.**
> > >
> > > Your mentioned papers define a path at a finer-grained granularity of neurons, whereas the path we mentioned here is at a coarser granularity of layers, i.e., one searchable layer has several parallelled candidate operations/paths and each operation can be viewed as one path between the layer input and layer output. You could refer to the visualization in Figure 1 of DARTS (H. Liu, ICLR’19), which uses the weighted sum of all candidate operations as the output of each layer, i.e., activates all the paths.
> > >
> > > **2. Discussions for the accelerator search algorithm.**
> > >
> > > **General applicability:**
> > > Our search approach is general and can be applicable to different hardware architectures, because our approach described in Sec. 3.3 does not hold any prior assumptions about the adopted hardware architecture. Specifically, for any target hardware architecture, including TPU-like or GEMM or other accelerators, our search approach can be directly applied once given (1) a simulator to estimate the hardware cost, and (2) a set of user-defined searchable design knobs (e.g., algorithm-to-architecture mapping methods, the number of memory hierarchy, buffer size allocation, etc) abstracted from the target hardware architecture. Furthermore, the abstracted design knobs from our template, such as PE numbers and buffer size allocation, also exist in most existing hardware architectures. As such, the experiment results in this work can provide useful insights to efficient accelerator designs when considering other architectures and to largely speed up the development of TRIPS on top of other hardware templates.
> > >
> > > **Workload of extending our search approach to other accelerators:**
> > > Extending our search approach to other accelerators requires the following steps: (1) Extracting the user-interested design knobs from the target accelerator template; (2) Calibrating the hardware cost model which receives the specified design knobs and returns the estimated hardware cost, and (3) Wrapping up with our provided search codes. In particular, Step (1) and (3) are quite straightforward, and Step (2) may be the most time- and labor-consuming part, which can be largely simplified by taking advantage of state-of-the-art estimation tools of accelerator performance, such as the Timeloop and Accelergy mentioned in Sec. 4.1. Therefore, you can see that overall it’s convenient to adapt our search approach to different accelerators.
> > > We will release TRIPS to the public upon acceptance, for which we will add detailed instructions and examples in our github repo to show how TRIPS can be used when considering other templates such as Eyeriss.

---

> > > > ### Comment · AnonReviewer2 · 2020-11-23
> > > > **Fair enough**
> > > >
> > > > Please add the discussion of path and the comparison with prior work on path to the paper.
> > > >
> > > > Please also consider adding a few sentences talking about the general applicability of the tool. It doesn't have to be as comprehensive as what's here, but a few sentences that reinforces the general applicability would be good.

---

> > > > > ### Author Response · Authors · 2020-11-23
> > > > > **Further response to Reviewer 2**
> > > > >
> > > > > Thanks for the valuable suggestions and we believe the discussions will strengthen our work. We will discuss the two points in the final version.

---

### Decision · Program_Chairs · 2021-01-07
**Final Decision**

**Decision:**

Reject

**Comment:**

This paper introduces a methodology for jointly optimizing neural network architecture, quantization policy, and hardware architecture. There are two key ideas:
- Heterogeneous sampling strategy to tackle the dilemma between exploding memory cost and biased search.
- Integrates a differentiable hardware search engine to support co-search in a differentiable manner.

The paper tackles an important research problem and experimental results are good.

There are two related issues with this paper:
1. Comparison to one-shot NAS: one-shot NAS methods only need to train the super-net once and then can be applied to multiple use-cases, while the proposed methodology needs to be executed for each use-case.
2. It is not clear whether differentiable search is needed for this joint optimization problem modulo existing tools.

Overall, my assessment is that the paper is somewhat borderline and with some more work will be ready for publication.